# ERK Inhibitor Ulixertinib Inhibits High-Risk Neuroblastoma Growth In Vitro and In Vivo

**DOI:** 10.3390/cancers14225534

**Published:** 2022-11-10

**Authors:** Yang Yu, Yanling Zhao, Jongmin Choi, Zhongcheng Shi, Linjie Guo, John Elizarraras, Andy Gu, Feng Cheng, Yanxin Pei, Dai Lu, Muller Fabbri, Saurabh Agarwal, Chunchao Zhang, Sung Yun Jung, Jennifer H. Foster, Jianhua Yang

**Affiliations:** 1Center for Cancer and Immunology Research, Children’s National Research Institute, Children’s National Hospital, Washington, DC 20010, USA; 2Texas Children’s Hospital, Department of Pediatrics, Dan L. Duncan Cancer Center, Baylor College of Medicine, Houston, TX 77030, USA; 3Advanced Technology Cores/Office of Research, Baylor College of Medicine, Houston, TX 77030, USA; 4Department of Molecular and Cellular Biology, Baylor College of Medicine, Houston, TX 77003, USA; 5Department of Pediatrics, The George Washington University School of Medicine and Health Sciences, Washington, DC 20010, USA; 6Rangel College of Pharmacy, Texas A&M University, Kingsville, TX 78363, USA; 7Department of Pharmaceutical Sciences, College of Pharmacy and Health Sciences, St. John’s University, Queens, NY 11439, USA

**Keywords:** neuroblastoma, ulixertinib, ERK inhibition, combination therapy, c-Myc/N-Myc

## Abstract

**Simple Summary:**

Neuroblastoma (NB) is the most common extracranial solid tumor in children, and the majority of patients with high-risk disease are unable to be cured. There is an urgent need to design novel therapeutics for NB patients. In NB, the RAS-MAPK pathway plays a crucial role in essential processes such as cell proliferation, survival, and chemoresistance. In the present study, we determined the therapeutic potential of the ERK inhibitor ulixertinib in NB using a panel of NB cell lines, patient-derived xenograft (PDX) cell lines, and NB xenograft mouse models. Ulixertinib significantly and potently inhibited NB cell proliferation and tumor growth, as well as prolonged survival in the treated mice. Additionally, ulixertinib synergistically sensitized NB cells to the conventional chemotherapeutic drug doxorubicin. This study provides proof-of-concept pre-clinical evidence for exploring ulixertinib as a novel therapeutic approach for NB.

**Abstract:**

Neuroblastoma (NB) is a pediatric tumor of the peripheral nervous system. Approximately 80% of relapsed NB show RAS-MAPK pathway mutations that activate ERK, resulting in the promotion of cell proliferation and drug resistance. Ulixertinib, a first-in-class ERK-specific inhibitor, has shown promising antitumor activity in phase 1 clinical trials for advanced solid tumors. Here, we show that ulixertinib significantly and dose-dependently inhibits cell proliferation and colony formation in different NB cell lines, including PDX cells. Transcriptomic analysis revealed that ulixertinib extensively inhibits different oncogenic and neuronal developmental pathways, including EGFR, VEGF, WNT, MAPK, NGF, and NTRK1. The proteomic analysis further revealed that ulixertinib inhibits the cell cycle and promotes apoptosis in NB cells. Additionally, ulixertinib treatment significantly sensitized NB cells to the conventional chemotherapeutic agent doxorubicin. Furthermore, ulixertinib potently inhibited NB tumor growth and prolonged the overall survival of the treated mice in two different NB mice models. Our preclinical study demonstrates that ulixertinib, either as a single agent or in combination with current therapies, is a novel and practical therapeutic approach for NB.

## 1. Introduction

NB is the most common extracranial tumor in children, and accounts for 15% of childhood malignancy-related deaths [1]. Although low- and intermediate-risk NB is highly likely to be cured, high-risk NB usually recurs with incurable disease despite intensive treatment with multiple modalities [2]. High-risk NB can develop sustained drug resistance during chemotherapy, which contributes to relapsed disease [3]. Therefore, finding more effective targeted therapies that can reduce drug resistance has been the focus of NB cancer research for decades.

The MAPK pathway is the central signal cascade that promotes cell proliferation. This pathway can be activated by growth factors binding to the corresponding receptors, leading to the activation of the RAS, RAF, MEK, and ERK pathways [4]. ERK is a major effector kinase of the MAPK pathway that activates various substrates through phosphorylation and triggers multiple cellular responses, including cell proliferation, by upregulating cell cycle genes [5]. Overactivation of the MAPK pathway due to the mutations of many pathway-related components is observed in numerous cancers, including relapsed NB [6]. Therefore, targeting the ERK/MAPK pathway has been an attractive therapeutic strategy in treating NB.

In addition to promoting cell proliferation, aberrant activation of the Raf/MEK/ERK pathway has been demonstrated to induce drug resistance in cancer cells [7]. The hyperactivated ERK pathway can be suppressed by both BRAF and MEK inhibitors, which has shown impressive results thus far [8,9]. However, tumors in most patients develop resistance after about one year of treatment with these inhibitors, establishing ERK, the terminal master kinase, as an ideal target for small-molecule inhibitors [10,11,12]. It has been demonstrated that ERK activation and subsequent RSK activation suppress GSK-3β activity [13]. Active GSK-3β can promote MYC phosphorylation and degradation in an SCF FBW7-dependent manner, which may also apply to N-Myc destabilization [14]. ERK activation can also upregulate c-Myc and N-Myc in NB [15,16]. Inhibition of ERK activation may thus reduce c-Myc and N-Myc levels and destabilize both proteins. It has been known that increased levels of c-Myc or N-Myc result in the development of drug resistance in multiple cancers, such as NB [17], leukemia [18], endometrial cancer [19], hepatocellular carcinoma (HCC) [20], human small cell lung carcinoma [21], and pancreatic cancer [22]. Ulixertinib, a novel selective ERK inhibitor, has shown promising preclinical activities in treating several cancers in vitro and in clinical trials [23]. However, whether ulixertinib can effectively treat NB or reduce the chemoresistance of NB cells remains unclear.

The present study demonstrates that ulixertinib inhibits NB cell proliferation in vitro and NB tumor growth in vivo. We further delineated ulixertinib-induced transcriptomic and proteomic alterations in multiple signaling pathways in NB cells by using RNA-Seq and mass spectrometry analyses. Additionally, ulixertinib sensitizes NB cells to doxorubicin-induced apoptosis, suggesting ulixertinib as a novel and effective treatment for relapsed and chemo-resistant NB.

## 2. Materials and Methods

### 2.1. Reagents

Ulixertinib (BVD-523) (CT-VRT752) was purchased from ChemieTek (Indianapolis, IN, USA). Fetal bovine serum (FBS) (35-011-CV), RPMI-1640 medium (10-040-CV), penicillin/streptomycin (30-002-CI), and 0.25% trypsin (25-050-CI) were purchased from Corning Incorporated (Corning, NY, USA). Bovine serum albumin (BSA) (A7906), dimethyl sulfoxide (DMSO) (D8418), and 3-(4, 5-dimethylthiazol-yl)-2,5-diphenyltetrazolium bromide (MTT) (M5655), were purchased from Sigma-Aldrich (Saint Louis, MO, USA). Anti-c-Myc (9E10) (sc-40), anti-α-tubulin (10D8) (sc-53646), and anti-N-Myc (NCM II 100) (sc-56729) were purchased from Santa Cruz Biotechnology (Dallas, TX, USA). ERK1/2 (#9102L), phospho-RSK (#9341), RSK (#9355), Cleaved Caspase 3 (#9664), anti-rabbit (#7074S), and anti-mouse (#7076S) secondary antibodies were purchased from Cell Signaling Technology (Danvers, MA, USA).

### 2.2. NB Cell Lines and Patient-Derived Xenograft Cells

Nine human NB cell lines were used in this study, including five MYCN-amplified NB cell lines, LAN-1, NGP, SK-N-BE(2), CHLA136, IMR-32, and four MYCN-non-amplified NB cell lines, CHLA255, SH-SY5Y, SK-N-AS, and LAN-6. Human bone marrow stromal cell line HS-5 was purchased from ATCC. All cell lines were maintained in RPMI-1640 medium, supplemented with 20% (*v/v*) heat-inactivated fetal bovine serum (FBS), 100 units/mL penicillin, and 100 μg/mL streptomycin at 37 °C in 5% CO_2_. All cell lines were authenticated via short tandem repeat (STR) analysis. Mycoplasma testing was performed by the LookOut^®^ Mycoplasma PCR Detection Kit (MP0035, Sigma-Aldrich). Patient-derived xenograft (PDX) COG-N-519x, COG-N-564x, COG-N-618x, and COG-N-700x cells were obtained from the Children’s Oncology Group (COG) Cell Culture and Xenograft Repository (www.COGcell.org) (Accessed on 4 March 2022) and maintained in Iscove’s modified Dulbecco’s medium supplemented with 20% fetal bovine serum, 4 mM L-glutamine, 1X ITS (5 µg/mL insulin, 5 µg/mL transferrin, 5 ng/mL selenous acid) (Biotechne, AR013, Minneapolis, MN, USA).

### 2.3. Cell Proliferation Assay

NB cell lines, NB PDX cells, and HS-5 cells were seeded in 96-well plates (3000 cells per well) in culture media. After incubating overnight, ulixertinib was added at indicated concentrations for five days. Cell proliferation was measured using Cell Counting kit-8 (CK04-20) purchased from Dojindo Molecular Technologies Inc. Each experiment was performed in three replicates.

### 2.4. Anchorage-Independent Colony Formation Assay

The soft agar assay was performed as described previously [24]. Briefly, 0.5% agar in cell culture media suspension was plated in 6-well plates and allowed to solidify, followed by adding an upper layer of agar (0.3%) containing 3 × 10^3^ NB cells/well. Ulixertinib was added at the indicated concentrations to the soft agar surface the next day. After four weeks, the cells were stained with 500 μL of 0.5% (*w/v*) MTT solution. Images of the colonies were captured and counted using VersaDoc Imaging System (Bio-Rad). Each assay was performed in triplicate.

### 2.5. Immunoblotting

After administration of ulixertinib at the indicated concentration and time point, cells were harvested and lysed using RIPA buffer containing 50 mM Tris-HCl (pH 7.4), 1% NP-40 (IGEPAL CA-630) (#I8896, Sigma-Aldrich), 0.25% sodium deoxycholate, 150 mM NaCl, 1 mM EDTA, 0.1 mM sodium orthovanadate, 0.5 mM PMSF, 1 mM DTT, 10 µg/mL leupeptin, 10 µg/mL aprotinin, 1 mM benzamidine, and phosphatase inhibitor cocktail 2 and 3 (#P5726 and #P0044, Sigma). Total cell lysate (50 µg) was used for SDS-PAGE and transferred onto PVDF membranes (Millipore). After blocking with 5% non-fat milk, membranes were incubated with primary antibodies at 4°C overnight. HRP conjugated secondary antibodies and ECL Western blotting Kit (GE Health) were used for signal detection. All the whole western blot figures can be found in the Appendix A.

### 2.6. Synergy Studies

About 3 × 10^3^ cells were seeded in each well of 96-well plates for CHLA255, SK–N-BE(2), SK-N-AS, and NGP cells, then treated with ulixertinib, doxorubicin, or a combination of both drugs. The CI values and dose-reduction indices (DRIs) were calculated using the Chou–Talalay method for drug interactions, with CompuSyn software for the different fractions affected [25]. CI < 1, =1, and >1 indicate synergism, additive effect, and antagonism, respectively. DRI > 1 and <1 indicate a favorable and an unfavorable dose reduction, respectively.

### 2.7. NB Xenograft Mouse Model

The NB xenograft mouse model was established as described by other studies [26,27]. NSG mice were purchased from The Jackson Laboratory and maintained at the Baylor College of Medicine animal care facility. Eight- to twelve-week-old sex-matched mice were intravenously injected with 1 × 10^6^ NB cells with firefly luciferase expression (CHLA136-Fluc and CHLA255-Fluc). Two weeks post-injection, mice were divided into two groups and intraperitoneally injected with ulixertinib (50 mg/kg) or vehicle for three weeks. Mice were assessed by a Xenogen IVIS 100 instrument every week. Xenogen images of mice were taken 15 min after injection of 1.5 mg D-luciferin (#122799, Perkin Elmer) intraperitoneally. For all experiments, the xenogen exposure time was set at 3 min. Mouse survival time was defined as the length of time from tumor cell injection until the end of the study or to euthanasia due to severe symptoms caused by tumor progression (for example, over 15% weight loss, weakness, seizures, inability to eat or drink, inability to stand, immobility, and/or paralysis). Percent survival was determined using a Kaplan–Meier analysis and a log-rank (Mantel-Cox) test. All animal procedures were approved by the Institutional Animal Care and Use Committee of Baylor College of Medicine.

### 2.8. RNA-Seq Analysis

NGP cells were treated with ulixertinib at 7.5 µmol/L for 24 h, the same volume of DMSO was used as the control. RNA was harvested using TRIzol™ Reagent (Thermo Fisher, Waltham, MA, USA). About 100 ng of total RNA was used for the construction of sequencing libraries. RNA libraries for RNA-seq were prepared using SMARTER mRNA-Seq Library Prep Kit following the manufacturer’s protocols. RNA-Seq experiments were conducted by using Illumina paired-end sequencing technology. Raw reads were adaptor removed, and sequencing quality was assessed with the bioinformatics software FastQC (version 0.11.2) (Babraham Bioinformatics, Babraham Institute, Cambridge, UK). Sequence quality scores, sequence duplication, and adaptor content were evaluated to decide if further filtering should be applied before the genome mapping. Reads were mapped against the human reference genome (GRCh38.p13 assembly) using the aligner, HISAT2 (v2.1.0) (Lyda Hill Department of Bioinformatics, University of Texas Southwestern Medical Center, Dallas, TX, USA). The mapped reads were subsequently assembled into transcripts or genes using the assembler, StringTie (v1.3.5) (The Center for Computational Biology at Johns Hopkins University, Baltimore, MD, USA).

### 2.9. Mass Spectrometry Data Analysis

In order to determine proteomic changes in NB cells (NGP) with the ulixertinib treatment, three biological replicates of each condition were harvested after DMSO vehicle or ulixertinib treatment for 24 h. Harvested cells were washed with cold PBS and lysed with 10 sample volumes of 50 mM ammonium bicarbonate with 1 mM CaCl2. Cell suspensions were lysed and double-trypsinized as described previously [26]. Double-digested peptides were fractionated into 15 fractions and pooled into 5 fractions in the Stage-tip C18 column, as described previously [26]. Pooled fractions of peptide were enriched on a 2 cm trap column (100 µm i.d.) and separated by 5 cm analytical column (150 µm i.d.) containing Reprosil-Pur Basic C18 (1.9 µm, Dr. Maisch GmbH, Germany). A nanoLC-1200 (Thermo Scientific, Lenexa, KS, USA) delivered a 75 min discontinuous gradient of 4 to 24 % of acetonitrile/0.1% formic acid at a flow rate of 800 nL/min. A Orbitrap Fusion mass spectrometer (Thermo Scientific) was operated in data dependent acquisition mode in the following parameters, precursor was scanned by at 300–1400 *m*/*z* range, 120 k resolution at 400 *m*/*z*, AGC target of 5 × 105 (50 ms maximum injection time). Cycle time was top 3 s selected MS1 signal using Quadrupole filter in 2 *m*/*z* isolation window, 15 s exclusion time. The HCD fragmented MS2 ions were detected by ion trap with rapid scan, 5 × 10^3^ AGC target, and 35 ms of maximum injection time. Proteome Discoverer 2.1 software (Thermo Fisher) with the Mascot 2.4 search engine (Matrix Science, Chicago, IL, USA) was used for spectra analysis using a target-decoy Human RefSeq proteome database with a 1% false discovery rate (FDR) of percolator validation based on q-value. The acetylation of the N-terminus and oxidation of methionine were allowed for dynamic modification. A 20 ppm of precursor mass tolerance, 0.5 Da of fragment mass tolerance, and a maximum of two missed cleavages was allowed. A label-free quantification strategy (iBAQ) was adopted for identified proteins quantification calculated by the gpGrouper algorithm [27]. Protein abundances are median normalized across the experiments.

### 2.10. Statistical and Bioinformatics Analysis

Data were represented as the means ± standard error. All experiments were repeated at least three times. A two-tailed Student’s *t*-test was used to determine the statistical significance of in vitro assay between drug treatment and the control groups. For survival analysis, the statistical significance was determined using a log-rank (Mantel-Cox) test. Differential gene expression (DGE) analysis of RNA-Seq was conducted with the R statistical package DESeq2 (v1.28.1) (Genome Biology Unit, European Molecular Biology Laboratory, Heidelberg, Germany). Genes were considered significantly altered if they had more or less than two-fold changes (FC ≥ 2 or FC ≤ 0.5) with an adjusted *p*-value less than 0.05. Only coding genes were retained on the list. For DGE analysis of proteins from MS, an unpaired *t*-test was adopted and a *p*-value < 0.05 was considered statistically significant. We performed gene set analysis on DGEs by using an over-representation test on the ConsensusPathDB [28] web-portal. Genes or proteins detected in our experiments were selected on the background list. KEGG and Reactome pathway databases were considered. Differentially expressed proteins were submitted to the STRING [29] database (https://string-db.org/) to evaluate global protein–protein interaction networks. The STRING version is 11.5 (Accessed on 12 August 2021). All parameters are default settings but confident scores. We only keep interaction scores higher than 0.70 (high confidence). The interaction score is a combined scored calculated by combining the probabilities from the different source and scaled between 0 and 1.

## 3. Results

### 3.1. Ulixertinib Significantly Inhibits NB Cell Proliferation

To test whether ulixertinib can inhibit cell proliferation and promote apoptosis in human NB cells, we first treated a panel of NB cell lines and PDX cell lines, including five *MYCN*-amplified (LAN-1, IMR-32, NGP, SK-N-BE(2), and CHLA136), four *MYCN* non-amplified (CHLA255, SH-SY5Y, SK-N-AS, and LAN-6), four NB PDX *MYCN*-amplified (COG-N-519x), and *MYCN* non-amplified (COG-N-564x, COG-N-618x, and COG-N-700x) cell lines. Our data demonstrate that ulixertinib significantly inhibited cell proliferation of all NB and PDX cell lines tested in a dose-dependent manner regardless of the *MYCN* status (Figure 1A–C). Treatment of a noncancerous bone marrow stromal fibroblast cell line HS-5 with ulixertinib showed inhibition of cell proliferation at a very high drug dose, in contrast to NB cell lines (Figure 1A). Different cell lines displayed a wide range of half-maximal inhibitory concentrations (IC_50_), in which CHLA255 was most sensitive, and LAN-1 and COG-N-618x were found to be least sensitive to ulixertinib, respectively (Figure 1D–F). Furthermore, ulixertinib significantly and dose-dependently inhibited different NB cells’ anchorage-independent colony formation (Figure 1G). Ulixertinib at the respective IC_50_ concentrations dramatically reduced colony sizes and numbers compared to controls. Across all cell lines, several colonies were observed in higher concentrations (5-fold IC_50_) (Figure 1H–K). Our data show that ulixertinib can suppress NB cell proliferation and anchorage-independent colony formation capacity in vitro.

### 3.2. Ulixertinib Inhibits RSK Phosphorylation and Downregulates c-Myc/N-Myc Protein Levels in NB Cells

Ulixertinib has been shown to inhibit RSK phosphorylation induced by ERK activation in other cancer cell types [23]. We hypothesized that ulixertinib has the same effects on ERK signaling in NB cells. Ulixertinib treatment in all tested NB cells significantly inhibited RSK1 phosphorylation (Figure 2). It suggests that ulixertinib also blocks ERK activation in NB cells. *MYCN* and *MYC* play a crucial role in NB tumor development. The previous study shows that both are the downstream targets of ERK [15,16]. We then explored whether ulixertinib-mediated ERK inhibition affects c-Myc or N-Myc protein levels. Our data show that ulixertinib treatment, in a time-dependent manner, significantly down-regulated N-Myc protein levels in *MYCN* amplified cell lines (NGP, SK-N-BE(2), CHLA136, and LAN-1) (Figure 2A–D), and c-Myc protein levels in *MYCN* non-amplified cell lines (LAN-6 and CHLA255) (Figure 2E,F). Since MYCN is an important prognostic factor in NB and is known to drive NB tumorigenesis and progression, our data on ulixertinib-mediated N-Myc inhibition highlight the potential of ulixertinib as a novel therapeutic approach for high-risk NB.

### 3.3. Ulixertinib Induces Extensive Transcriptomic Changes in NB

To uncover how ulixertinib inhibits cell proliferation in NB, we performed a genome-wide transcriptomic analysis with RNA-Seq in ulixertinib-treated NB cells (Figure 3A). This analysis revealed 120 upregulated and 187 downregulated genes in response to ulixertinib treatment compared to the controls (Figure 3A and Appendix A). Gene set enrichment analysis by the over-representation test has revealed inhibition of multiple pathways and their cross-talks (Figure 3B,C). As expected, MAPK signaling was significantly decreased as well as EGFR, VEGF, and WNT signaling pathways, demonstrated by multiple shared key components (Figure 3C, left panel). Overexpression and dominance of these survival pathways drive NB tumorigenesis and the malignant transformation [30,31]. NGF-stimulated transcription and NTRK1 signaling (Figure 3C, right panel) are the other two pathways significantly inhibited by ulixertinib, both of which are required for neuronal growth or nervous system development [32]. Among the upregulated genes, MBNL2 [33], SEPT4 [34], REPS2 [35], OTUD5 [36], and TXNIP [37] were shown to play a tumor-suppressive role in many tumors. To determine the correlation between these upregulated genes with NB patient prognosis, we analyzed NB patient datasets in the R2 database (https://hgserver1.amc.nl/cgi-bin/r2/main.cgi, access on 1 April 2021) and found that decreased expression of these genes predicted a significantly poor outcome (Appendix A). These results suggest that these genes upregulated by ulixertinib treatment exert a tumor-suppressive function in NB cells.

### 3.4. Ulixertinib Exerts Extensive Proteomic Changes in NB

To further delineate the mechanisms of ulixertinib-mediated ERK pathway inhibition in NB and to determine the proteomic effects of ulixertinib in NB, we performed mass spectrometry analysis in NB cells (NGP) with and without ulixertinib treatment. Our proteomic analysis data identified a total of 157 differentially expressed proteins out of a total of 6105 measurable proteins (Appendix A), with a threshold of *p* < 0.05 and a log fold change of expression with an absolute value of at least 1.0 (Figure 3D). Ulixertinib treatment led to the upregulation of 72 proteins and the downregulation of 85 proteins. The complete lists of gene names are shown in Appendix A. In our analysis, ulixertinib is found to significantly inhibit cell cycles or cell cycle-related pathways and DNA replication/synthesis pathways, as revealed by gene set analysis using an over-representation test (Figure 3E). STRING analysis (https://string-db.org/, access on 12 August 2021) of ulixertinib-inhibited proteins revealed global protein–protein interaction networks and core interactomes (Figure 3F). The core components are ATAD2, AURKB, CENPF, CENPM, CDCA5, FAM64A, KIF20A, KIF22, KIFC1, KIF2C, KPNA2, MK167, NUSAP1, PRC1, RRM2, SPDL1, TOP2A, TPX2, UBE2T, and UHRF1. Among these proteins, MK167 is a cellular marker for proliferation [38]. KIF (kinesin superfamily) proteins (KIF20A, KIF22, KIFC1, and KIF2C) are microtubule-dependent molecular motors that play vital roles in cellular transport and cell division [39]. Moreover, high expression of some of these identified proteins inhibited by ulixertinib, including CDCA5, ATAD2, CDC25A, CDK2, ANAPC10, RRM2, LRG5, SKP2, CENPM, and UBE2T, are strongly associated with worse overall survival of NB patients (Appendix A). Ulixertinib treatment enhanced the PRUNE2 level (Appendix A), and PRUNE2 has been identified as a pro-apoptotic effector in NB [40]. Decreased expression of PRUNE2 predicted poor NB patients’ outcomes (Appendix A). Our proteomic data indicate that ulixertinib inhibits cell proliferation and cell cycle by regulating the expression levels of multiple factors involved in NB development.

### 3.5. Ulixertinib Sensitizes NB to Chemotherapeutic Agent Doxorubicin

Doxorubicin, a chemotherapy agent standardly used in the treatment of high-risk NB, exhibited stronger cytotoxicity in combination with Raf inhibition in NB cells [41]. To determine whether blocking the ERK pathway using ulixertinib could enhance doxorubicin-induced apoptosis in NB cells, we treated four NB cell lines, including SK-N-AS, NGP, SK-N-BE(2), and CHLA255, with a combination of doxorubicin and ulixertinib and determined the combination index (CI) using the Chou–Talalay method. CI values < 1 signify synergism, =1, additive effects, and >1, antagonism [42]. A strong synergy, which was evaluated by different effective doses (ED: ED50, ED75, ED90, and ED95), was observed in the combination of doxorubicin and ulixertinib in all four NB cell lines (CI < 0.72 at ED50; CI < 0.19 at ED95) (Figure 4A–D). In addition, enhanced apoptosis in the cells with the treatment combination of ulixertinib and doxorubicin was observed using an immunoblotting assay to assess cleaved Caspase 3, an apoptosis marker, compared to single agent treatment (Figure 4E,F). To further confirm the results, apoptosis of different treatment groups was determined by flow cytometry detection of their sub-G0 DNA contents using propidium iodine (PI) staining. Results showed that ulixertinib and doxorubicin combination treatment increased the sub-G0 population compared to ulixertinib or doxorubicin treatment alone (Appendix A). Our data indicate that ulixertinib significantly sensitizes NB cells to doxorubicin-induced apoptosis and could serve as a promising therapeutic agent for treating chemo-resistant NB when combined with conventional chemotherapeutic agents.

### 3.6. Ulixertinib Inhibits NB Tumor Growth in Xenograft Mouse Models

To further evaluate the therapeutic potential of ulixertinib in treating NB, we tested its effect on tumor growth using two xenograft NB mouse models. We used CHLA136-Fluc (*MYCN* amplified) and CHLA255-Fluc (c-Myc overexpressed) to generate these xenograft mouse NB tumor models. Randomized mice were grouped in different cohorts and treated with ulixertinib or vehicle daily for three weeks (Figure 5A and Figure 6A). Ulixertinib daily injection was well tolerated at a concentration of 50 mg/kg without apparent bodyweight loss and other adverse effects. We found that treatment with ulixertinib significantly inhibited the overall NB tumor growth compared to vehicle (Figure 5B,C and Figure 6B,C). We also found that ulixertinib-treated xenograft NB mice survived significantly longer when compared to control mice (Figure 5D and Figure 6D). Overall, this data clearly demonstrate the potency and efficacy of ulixertinib in inhibiting NB tumor growth and prolonging overall survival.

## 4. Discussion

Aberrant RAS-MAPK-ERK activity can lead to uncontrolled cell proliferation, immortalization, and tumorigenesis. In NB, activation of the RAS-MAPK-ERK pathway has been associated with enhanced tumor cell proliferation and chemoresistance, possibly contributing to the recurrent disease that more than half of NB patients experience [6]. Targeting the RAS-MAPK-ERK pathway, especially ERK, provides a promising solution for improving survival in high-risk NB. In this study, we demonstrated that ulixertinib, a specific ERK inhibitor, effectively inhibits NB cell proliferation, promotes apoptosis, and sensitizes NB to doxorubicin, partially through downregulating c-Myc/N-Myc and other key signaling pathways. For the first time, we showed that in preclinical models ulixertinib inhibits NB tumor growth. We further delineated the underlying mechanisms of ulixertinib as a therapeutic agent for NB through the analysis of transcriptomics and proteomics. This could provide a solid foundation and clinical application of ulixertinib for use in the treatment of NB, especially for patients with refractory or recurrent disease.

Targeting ERK using small molecules has been demonstrated to effectively inhibit the Ras/Raf/MEK/ERK pathway in several models of cancers, including NB. Five ERK inhibitors have been tested in clinical trials, but only GDC-0994 and ulixertinib have shown antitumor activity [43]. Compared to GDC-0994, ulixertinib has exhibited anticancer activities for broader cancer types. More importantly, in clinical settings, cancer patients tolerated much higher doses of ulixertinib than GDC-0994 (1200 mg vs. 120–400 mg daily) [11,23,44,45]. Our study illustrates the efficacy of ulixertinib in treating NB in vitro and in vivo. Therefore, ulixertinib may be an ideal therapeutic agent to treat NB.

ERK inactivation by ulixertinib has been demonstrated in multiple cancers, including pancreatic cancer [46], melanoma, colon cancer [23], and glioma [47]. In these cancer cells, ulixertinib treatment significantly suppressed the phosphorylation of its target RSK [46]. Our study consistently revealed the same results in all tested NB cells. Various ERK inhibitors have been shown to have different effects on ERK phosphorylation. For instance, ERK phosphorylation was increased with ulixertinib; this was not observed with SCH772984 (an ERK inhibitor) in pancreatic cancer cells [46]. However, RSK phosphorylation is constantly suppressed by all ERK inhibitors tested, suggesting that RSK phosphorylation should be a reliable ERK activation marker regardless of cell types and inhibitors in NB.

In addition to blocking ERK activity, ulixertinib treatment also inhibited the Wnt/β-catenin pathway by downregulating LGR5 transcription (Figure 3A and Appendix A). Tumor samples from high-risk NB patients with and without *MYCN* amplification show the expression of canonical Wnt pathway target genes at high levels, indicative of Wnt pathway deregulation in NB [31]. High expression of LGR5 is reported in human colorectal adenomas and cancers [48], hepatocellular carcinoma, and basal cell carcinoma [49,50]. LGR5 is highly expressed in NB, which modulates Wnt signaling and is associated with increased proliferation [51]. Importantly, LGR5 also regulates MEK/ERK and Akt pro-survival signaling, pathways that are frequently activated in primary NBs [51]. This suggests that LGR5 could be one of the key components in ulixertinib-induced inhibition of pro-survival signaling. It has been well documented that overexpression of MYC or MYCN promotes NB cell proliferation [17]. Ulixertinib-induced ERK inhibition can efficiently inhibit NB cell proliferation by modulating various downstream targets, including c-Myc/N-Myc, required for the transcription of critical cell cycle genes. Marco Ciró et al. showed that ATAD2 is highly expressed in multiple types of human cancers and cooperates with MYC to activate MYC target genes transcriptionally [52]. Depletion of ATAD2 could reduce the mRNA expression of Myc and the key genes in hepatocellular carcinoma [53]. Our study demonstrates that ulixertinib downregulated ATAD2, ensuring an effective blocking of MYC/MYCN transcriptional activation of target oncogenes in NB cells. The exact roles of ATAD2 in NB development need to be further explored, given high ATAD2 expression is closely associated with worse survival outcomes. AURKB, one of the core components of ulixertinib-inhibited global protein–protein interaction networks (Figure 3F and Appendix A), promotes MYC protein stability by regulating its serine 67 phosphorylation [54]. Taken together, our findings indicate that ulixertinib might decrease MYCN/MYC levels by downregulating the expression of ATAD2 and AURKB.

Another possible mechanism of ulixertinib suppressing cell proliferation could be through RSK kinase inactivation-mediate p27^KiP1^ nuclear translocation, which can further inhibit the activation of cyclin A/E-CDK2 complexes that are required for G1 progression [55]. In addition to reduced activation of cyclin A/E-CDK2 complexes, the mass spectrometry results revealed that ulixertinib treatment reduces CDK2 protein levels, further suppressing G1 progression. Moreover, ulixertinib also reduces the levels of other cell cycle proteins, such as CDCA5, ANAPC10, SKP2, and CENPM. Therefore, ulixertinib may inhibit cell cycle progression by downregulating multiple key regulators. In human cervical carcinoma cells, downregulation of CDK2 has been shown to induce tumor cell cycle arrest and cell apoptosis [56]. We therefore expect that inhibition of CDK2 might also be one mechanism by which ulixertinib promotes apoptosis in NB cells.

Transcriptomic and proteomics analyses also revealed some upregulated genes following ulixertinib treatment in NB cells, such as TXNIP, MBNL2, and SEPT4. TXNIP is considered to be a potential tumor suppressor gene in multiple tumors, including breast cancer [57], hepatocellular carcinoma [58], non-small-cell lung cancer [59], and renal cell carcinoma [60]. The expression of TXNIP is at a low level in these tumor types, and the overexpression of TXNIP inhibits the proliferation of cancer cells. Lee YH, et al. found that the expression of MBNL2 was lost in the late stage of hepatocellular carcinoma development [33]. MBNL2-positive correlates with better 5-year overall survival [33]. Overexpression of MBNL2 inhibits cell proliferation, migration in vitro, and tumor growth in vivo [33]. Genetic evidence shows that SEPT4 suppresses tumor development by antagonizing the function of inhibitor of apoptosis proteins (IAPs) [34]. Consistently, our findings indicate that patients with elevated expression of these three genes had significantly longer overall survival compared to patients bearing tumors with low levels (Appendix A). The exact function of these genes in NB development requires further exploration.

Chemoresistance has long been a challenge in treating high-risk NB. Among ERK downstream targets, c-Myc and N-Myc have been demonstrated to promote chemoresistance of NB cells [61,62]. In our study, ulixertinib inhibits c-Myc/N-Myc levels in NB cells regardless of *TP53* status and *MYCN* status. This further enhanced doxorubicin sensitivity on all tested NB cells. Moreover, in addition to small molecules, RNA silencing of *MYCN* has been demonstrated to restore doxorubicin sensitivity in NB cells [63]. Enhancing the effectiveness of doxorubicin, a key component of high-risk NB induction chemotherapy, may help to improve end of induction disease response and thus potentially survival as well. Compared to the RNA silencing approach, treatments with small molecules are more efficient and controllable in in vivo studies and clinical applications. We also found that ulixertinib significantly reduced the levels of other promoters of chemoresistance, such as RRM2 and UBE2T. RRM2, a downstream target of ERK [64] and positive regulator of BCL2 [65], is a key determinant in suppressing apoptosis and promoting chemoresistance. In addition, high RRM2 expression has also been demonstrated to promote epithelial-mesenchymal transition (EMT) and angiogenesis in prostate cancer, resulting in poor patient outcomes [66]. So far, UBE2T has not been linked to being a downstream target of ERK signaling. Therefore, we speculate that ulixertinib could not only enhance the sensitivity of NB cells to chemotherapeutic drugs but may also suppress EMT and angiogenesis in NB. We chose to study doxorubicin given the existing pre-clinical data in combination with RAF inhibitors. One disadvantage of doxorubicin however is that it is not standardly used for patients with relapsed disease. Future studies can explore the impact ulixertinib has on chemotherapy regimens typically used for patients with relapsed disease. In the present study, treatment with ulixertinib induced the expression of PRUNE2 in NB cells (Appendix A). PRUNE2 is associated with a favorable prognosis in NB, and is found in the cytoplasm of favorable NB cells but not in unfavorable ones with *MYCN* amplification [67]. PRUNE2, triggered by DNA damage, also exerts a pro-apoptotic role by interacting with BCL2 and suppressing AKT pathway activity in NB [40,68]. A recent study shows that EPO, NGF, and HGF signaling pathways are upregulated in NB patients with no or partial response to chemotherapy [69]. Chemical inhibitors potentiate strong ERK signaling activation by EPO and NGF, providing a protective effect to NB cells [69]. Our study suggests that ulixertinib also inhibits the NGF-induced signaling pathway (Figure 3C). However, we also found the tumor relapse after treatment with ulixertinib alone for 3 weeks. This suggests that ulixertinib has to be used in combination with current chemotherapy to treat NB. Valencia-Sama I. et al. also reported that ulixertinib was synergistic and showed reversed resistance to SHP2 inhibition in neuroblastoma in vitro and in vivo [70]. Considering that a phase 2 clinical trial using ulixertinib to treat solid tumors, including NB, is in progress (NCT03698994), ulixertinib could be closer to clinical use compared to other therapeutic strategies for treating NB. However, more NB clinical trials using the combination of ulixertinib and chemotherapeutic drugs are needed to verify the efficacy of this novel regime.

## 5. Conclusions

Our study evaluated the therapeutic potential of ulixertinib-based treatment in NB. Ulixertinib significantly inhibits cell proliferation in NB cell lines and PDX cells in vitro. Moreover, ulixertinib inhibits tumor growth and prolongs the overall survival in preclinical tumor models. Transcriptomics and proteomic analysis revealed tumor-suppressive properties of ulixertinib on the expression of genes involved in NB development. Importantly, we also demonstrated the efficacy of ulixertinib in sensitizing NB cells to standard chemotherapy. Ulixertinib has the potential either as a single agent or in combination with standard chemotherapy to improve survival for patients with high-risk NB.

## Figures and Tables

**Figure 1 cancers-14-05534-f001:**
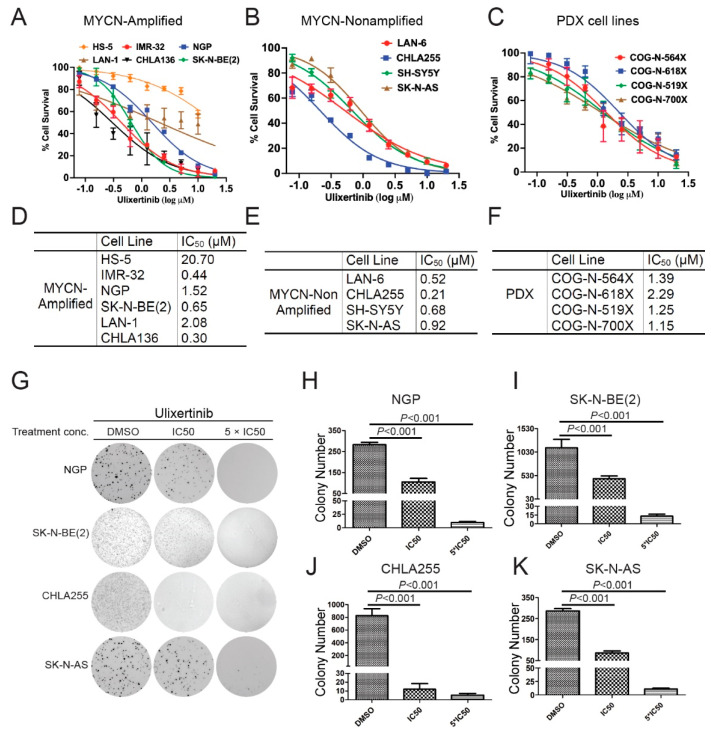
Ulixertinib inhibits NB cell proliferation and anchorage-independent growth in vitro. (**A**) Five MYCN amplified NB cell lines (IMR-32, NGP, SK-N-BE(2), LAN-1, and CHLA136) and a noncancerous bone marrow stromal fibroblast cell line, HS-5, were treated with the indicated concentrations of ulixertinib for 5 days. Cell proliferation was measured by CCK-8 assays. (**B**,**C**) Four MYNC- non-amplified NB cell lines (LAN-6, CHLA255, SH-SY5Y, and SK-N-AS) (**B**) and four PDX cell lines (COG-N-519x, COG-N-564x, COG-N-618x, and COG-N-700x) (**C**) were also treated with unlixertinib as in (A). Cell proliferation was measured by CCK-8 assays. (**D**–**F**) IC50 values were calculated based on the data collected in the CCK-8 assay from (**A**–**C**). (**G**) The effects of ulixertinib on anchorage-independent cell proliferation. Four NB cell lines (NGP, SK-N-BE(2), CHLA255, and SK-N-AS) were grown in soft agar with DMSO, 1 × IC50, and 5 × IC50 of ulixertinib for 4 weeks. Cells were stained with 0.5% (*w/v*) MTT solution to visualize colonies and photographed. (**H**–**K**) Colonies from (**G**) were counted in NGP (**H**), SK-N-BE(2) (**I**), CHLA255 (**J**), and SK-N-AS (**K**) cells. Colony numbers were represented as mean ± S.D. A two-tailed Student’s *t*-test was used to determine the statistical significance.

**Figure 2 cancers-14-05534-f002:**
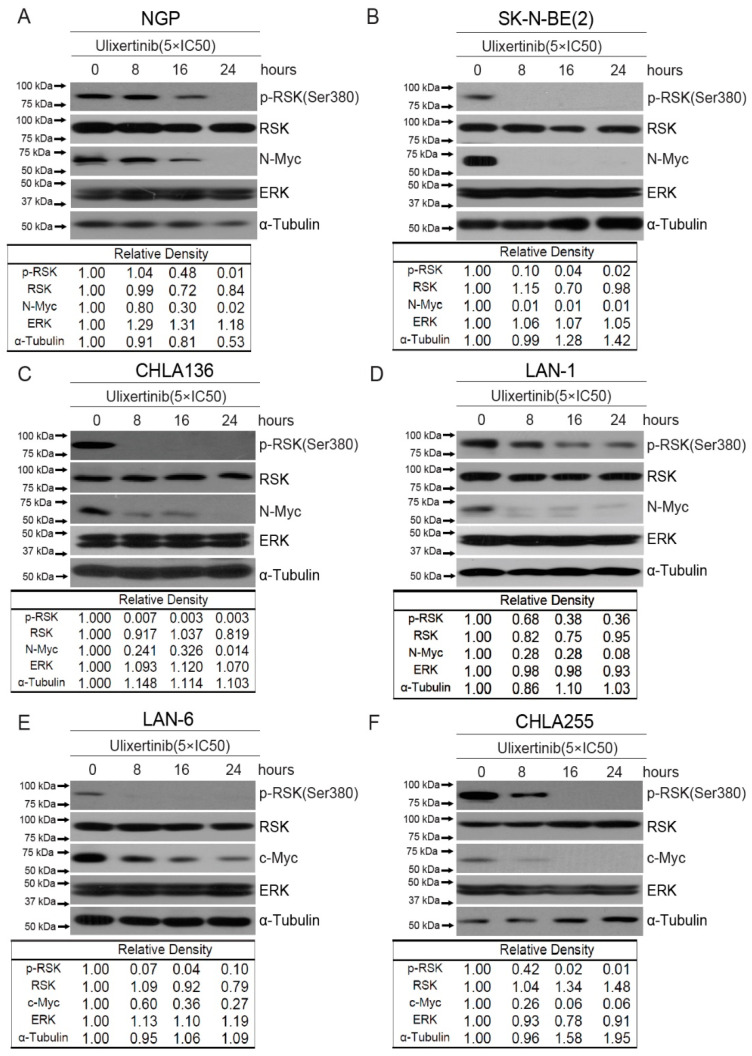
Ulixertinib suppresses ERK activation and reduces MYC/N-MYC levels in NB cells. Six NB cell lines, NGP (**A**), SK-N-BE(2) (**B**), CHLA136 (**C**), LAN-1 (**D**), LAN-6 (**E**), and CHLA255 (**F**) were treated with indicated doses of ulixertinib for variable periods (0 h, 8 h, 16 h, and 24 h). The proteins were extracted and subjected to SDS-PAGE for immunoblotting with the antibodies indicated. Gel densitometrical analysis was performed using ImageJ software (Version 1.52a, accessed on 23 April 2018) (Wayne Rasband, National Institutes of Health, USA).

**Figure 3 cancers-14-05534-f003:**
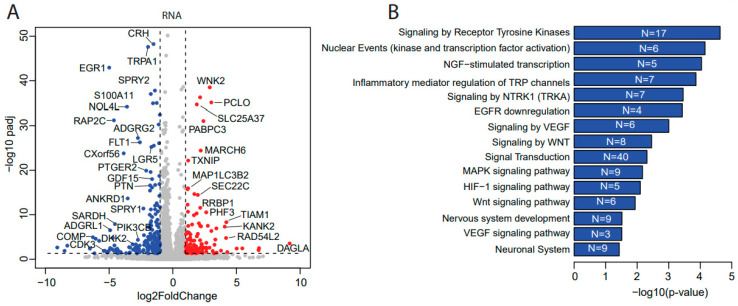
Transcriptomics and proteomics reveal ulixertinib-induced gene changes in NGP cells. (**A**) The differentially expressed genes revealed by DESeq2 analysis of RNA-Seq read counts. They are represented in the volcano plot in terms of their measured expression change (log2 of fold changes, *x*-axis) and the significance of the change (−log10 of adjusted *p* values, *y*-axis). The dotted lines represent the thresholds to define the differentially expressed genes. Red dots are up-regulated genes after the drug treatment (fold changes ≥ 2 and adjusted *p* < 0.05). Blue dots are down-regulated genes with the drug treatment (fold changes ≤ 0.5 and adjusted *p* < 0.05). (**B**) Top inhibited pathways revealed by down-regulated genes from RNA-Seq. N is number. (**C**) Key components inhibited and their associated pathways. (**D**) The differentially expressed proteins from MS were represented in the volcano plot in terms of their measured expression change (log2 of fold changes, *x*-axis) and the significance of the change (-log10 of *p* values, *y*-axis; unpaired *t*-test). The dotted lines represent the thresholds to define the differentially expressed proteins. Red dots are up-regulated proteins after the drug treatment (fold changes ≥ 2 and *p* < 0.05). Blue dots are down-regulated proteins with the drug treatment (fold changes ≤ 0.5 and *p* < 0.05). (**E**) Top inhibited pathways revealed by downregulated proteins from MS. N is number. (**F**) Protein–protein interaction networks analysis of down-regulated proteins from MS.

**Figure 4 cancers-14-05534-f004:**
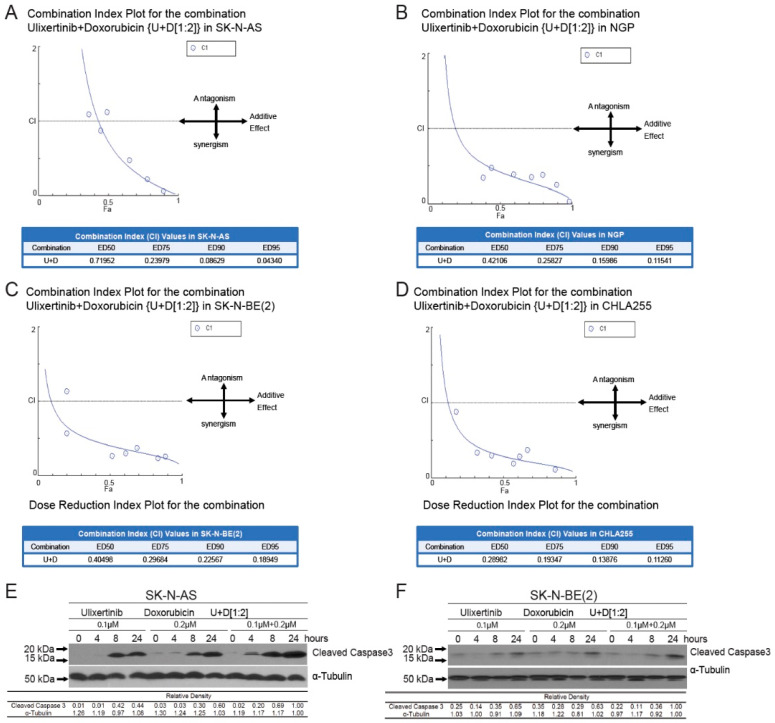
Ulixertinib enhances the anti-tumor activity of doxorubicin in human neuroblastoma cells. (**A**–**D**) CI plots for SK-N-AS, NGP, CHLA255, and SK-N-BE (2). Four NB cell lines, including SK-N-AS (**A**), NGP (**B**), SK-N-BE(2) (**C**), and CHLA255 (**D**) cells, were treated with ulixertinib, doxorubicin, and their combinations at a ratio of 1:2 respectively, for 72 h. The combination index (CI) values were calculated by the Chou–Talalay method for drug interactions using Compusyn software. Values of CI < 1, =1, and >1 indicate synergism, additive effects, and antagonism, respectively. The obtained CI values for the combination of ulixertinib and doxorubicin at different effective doses (ED50, ED75, ED90, and ED95). (**E**,**F**) SK-N-AS (**E**) and SK-N-BE(2) (**F**) cells were treated with either doxorubicin alone, ulixertinib alone, or their combinations for 4 h, 8 h, and 24 h. Then the whole-cell lysates were subjected to SDS-PAGE and immunoblotted with the cleaved-caspase 3 antibody. α-tubulin was used as a loading control in all samples. Gel densitometrical analysis was performed using ImageJ software (https://imagej.nih.gov/ij/index.html, accessed on 25 October 2022).

**Figure 5 cancers-14-05534-f005:**
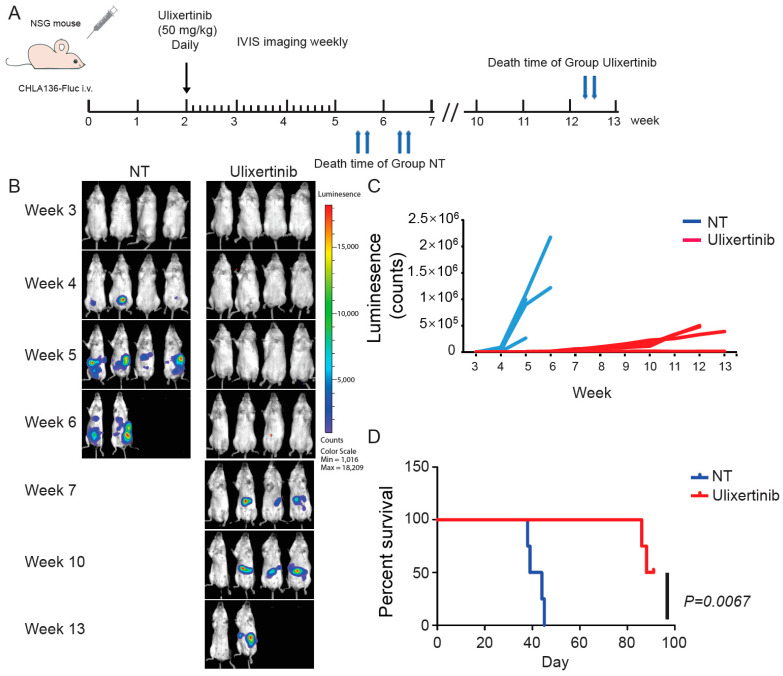
Ulixertinib inhibits tumor growth in the CHLA136-Fluc xenograft NB mouse model. (**A**) Treatment strategy of the xenograft NB mouse model. (**B**) Bioluminescent imaging photos of CHLA136-Fluc xenograft NB mice from the DMSO control group and the ulixertinib-treated group (50 mg/kg). (**C**) The values of bioluminescent imaging of xenograft NB mice were monitored during tumor growth. (**D**) The survival rate of CHLA136-Fluc xenograft NB mice treated with ulixertinib and control. Statistical analysis was performed by a Log-rank test. (χ^2^ = 7.344; *p* = 0.0067).

**Figure 6 cancers-14-05534-f006:**
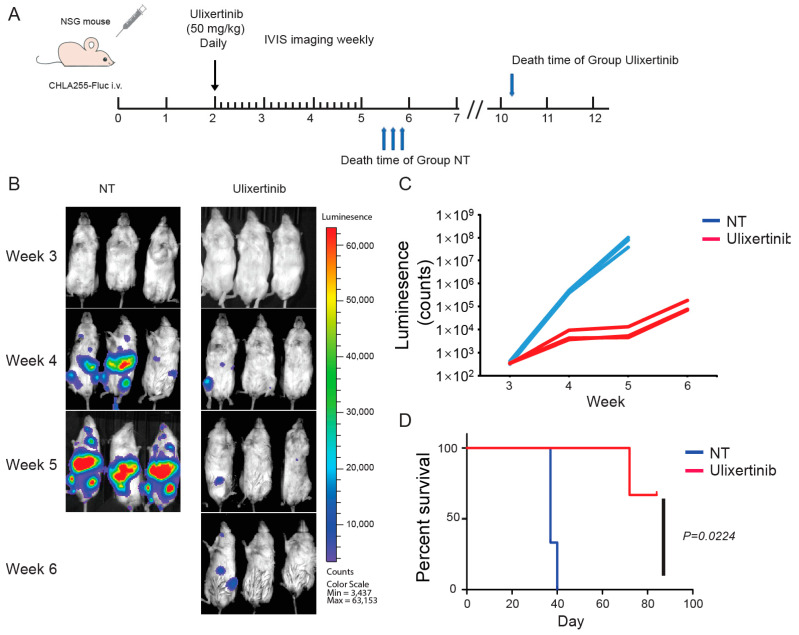
Ulixertinib inhibits tumor growth in the CHLA255-Fluc xenograft NB mouse model. (**A**) Treatment strategy of the xenograft NB mouse model. (**B**) Bioluminescent imaging photos of CHLA255 xenograft NB mice from the DMSO control group and the ulixertinib treated group (50 mg/kg). (**C**) The values of bioluminescent imaging of xenograft NB mice were monitored during tumor growth. (**D**) The survival rate of CHLA255 xenograft NB mice treated with ulixertinib and control. Statistical analysis was performed by a Log-rank test. (χ^2^ = 5.213; *p* = 0.0224).

## Data Availability

The mass spectrometry data for global proteomics profiling have been deposited to MASSIVE repository (MassIVE MSV000087511). RNA-Seq raw files are deposited onto NCBI Sequence Read Archive (SRA) database and the GEO accession number is GSE213153.

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
