# Peer review of "ERK Inhibitor Ulixertinib Inhibits High-Risk Neuroblastoma Growth In Vitro and In Vivo"

_cancers, 2022, doi:10.3390/cancers14225534_

Round 1

Reviewer 1 Report

Overall the authors have presented a comprehensive manuscript. I have noted some minor points, the authors should consider these.

1. Please mention how the cells were treated before RNA seq, In fig. 3b legend, please expand "N". 

2. the authors should comment/show on how the signaling regulation in cell lines at IC50 dose (fig. 2).

3. The authors should highlight the toxicity issues of Ulixertinib treated mouse. 50mg/kg bw per day may be a point of concern for future stuides. The authors should also comment on the tumor relapse after week 7.

4. The author should comment on combining the drug candidate with doxorubicin, what will be the advantages and disadvantages.

5. It is not clear from the methods that how the authors initially assesed the growth of the tumor, as in Week 3 there are no visible luminescence, please explain.

Author Response

We would like to thank the reviewers as well for their valuable comments. Below is a detailed point-to-point response to each of the reviewers’ comments.

Reviewer 1

  1. Please mention how the cells were treated before RNA seq, In fig. 3b legend, please expand "N". 

Thanks for your comment. We added detailed information about cell treatment for RNA-seq in section 2.8 RNA-Seq analysis. N in fig 3 legends is defined as number.

  1. the authors should comment/show on how the signaling regulation in cell lines at IC50 dose (fig. 2).

Thank you for your comment. We added some descriptions of the effect of ulixertinib on signaling in NB cells in Section 3.2.

  1. The authors should highlight the toxicity issues of Ulixertinib treated mouse. 50mg/kg bw per day may be a point of concern for future stuides. The authors should also comment on the tumor relapse after week 7.

Thank you. Based on your valuable comments, we re-edited Section 3.6 Ulixertinib inhibits NB tumor growth in xenograft mouse models. Tumor relapse after week 7 suggests that Ulixertinib has to be used in combination with current chemotherapy to achieve better therapeutic efficacy.

  1. The author should comment on combining the drug candidate with doxorubicin, what will be the advantages and disadvantages.

Thank you for your comment. We have now added the advantages and disadvantages of doxorubicin in combination experiments in discussion.

  1. It is not clear from the methods that how the authors initially assessed the growth of the tumor, as in Week 3 there are no visible luminescence, please explain.

Thank you for your comment. The tumor cannot be detected at Week 3 by IVIS system. This xenograft mouse model was used by many other groups (Clin Cancer Res. 2019, PMID: 31484667; Nat Commun. 2021, PMID: 33479234). The model has 100% penetrance when more than 106 NB cells are inoculated intravenously. In our study, sex- and age-matched mice were used in each experiment. The tumor growth and spread took more than 3 weeks and that’s the reason for less or no visible luminescence at week 3.

Reviewer 2 Report

Line33 Description of PDx should be given for first instance of the abbreviation

Line36 preclinical evidence for translation, (clinical translation is implied upon using preclinical evidence)

Fig.1A Graph legends are not legible and table should be clear

Fig1B The x axis of the colony number graphs are cropped out

Both the panels A and B should have the graphs and images numbered for explaining clearly in the legend. The figure panels needs to be edited/redone.

P values should indicate numbers, not asterisk.

Suggestion: In order to check survival assays post Ulixertinib treatment AnnexinV/PI staining and floe cytometry can be performed to understand apoptosis patterns in various cell lines.

Fig.2 panels should be numbered. The antibodies probed must have the size in kDa mentioned, the same way it is indicated in the supplementary panel.

Line267 Result heading 3.3 should be in the next page

Fig. 4 panels A and B needs numbering

FIg.5 A Schematic of the treatment strategy is unclear. The points representing week and days are misleading

The xenograft tumor model is described very briefly Line 365,366 without   references.

Line 516- synergistically with chemotherapy agents

Author Response

We would like to thank the reviewers as well for their valuable comments. Below is a detailed point-to-point response to each of the reviewers’ comments.

 Reviewer 2

Line33 Description of PDX should be given for first instance of the abbreviation

Thanks for your comment. We revised accordingly.

Line36 preclinical evidence for translation, (clinical translation is implied upon using preclinical evidence)

Thanks. We revised this sentence.

Fig.1A Graph legends are not legible and table should be clear, Fig1B The x axis of the colony number graphs are cropped out.

Thanks for your comment. We have revised Figure 1 based on your suggestion.

Both the panels A and B should have the graphs and images numbered for explaining clearly in the legend. The figure panels needs to be edited/redone.

P values should indicate numbers, not asterisk.

Thanks for your comment. We have revised it accordingly.

Suggestion: In order to check survival assays post Ulixertinib treatment AnnexinV/PI staining and floe cytometry can be performed to understand apoptosis patterns in various cell lines.

Thank you for your valuable suggestions. PI staining and Flow cytometry were used to examine the effect of ulixertinib, doxorubicin, and their combination on cell cycle distribution in human NB cell lines, SK-N-AS and SK-N-BE(2). We found that the sub-G0 DNA contents were significantly increased in ulixertinib and doxorubicin combination groups compared to ulixertinib or doxorubicin groups (Figure S3). We also found that ulixertinib induces NB cell apoptosis at a concentration of 0.5 uM for 24 hours. In futural works, we will examine the effect of ulixertinib on other NB cells using this assay.   

Fig.2 panels should be numbered. The antibodies probed must have the size in kDa mentioned, the same way it is indicated in the supplementary panel.

Thank you. Figure 2 has been revised based on your comments

Line267 Result heading 3.3 should be in the next page

Thank you. We have revised accordingly.

Fig. 4 panels A and B needs numbering

Thank you. Figure 4 has been revised based on your comments

FIg.5 A Schematic of the treatment strategy is unclear. The points representing week and days are misleading

Thank you. We have revised accordingly.

The xenograft tumor model is described very briefly Line 365,366 without references.

Thank you. Based on your comments, we re-edited the section “2.7. NB xenograft mouse model”. And references were added to the revised manuscript.

Line 516- synergistically with chemotherapy agents

Thank you. We have revised accordingly.

Reviewer 3 Report

Specific Comments:

           This manuscript uses the in vitro and in vivo approach to suggest the therapeutic potential of Ulixertinib-based treatment blastoma.

Strengths:

Well written manuscript
Data clearly shown in figures and images
Good level of interpretation

Limitations:

Sample size unknown in some places.

Please provide the catalog and company name for all chemicals/kits/reagents.

Please provide more detailed information on patients-derived xenograph.

It will be appropriate to describe the dosage and time of treatment in the material and method section. Please give a detailed description of how the experiment was done, what dosage was included, and what time point were considered for the study. It will help readers to reproduce the study for research purposes.

How many colonies were captured for counting?

For the immunoblotting section – Please include detailed information. What antibodies and concentrations were used with their catalog no and company name?

Please acknowledge the STRING site or link for the protein-protein interaction study.

Please correct lines 205-206 for the FC approach. It should be more or less than 2.0 FC for significantly altered genes.

Figure 1B lower bar diagram - The X-axis is not clear for both bottom figures. Please modify this.

Please provide the densitometric bar image for all the western blots.

Please give details about RNA seq data. What time point was considered for the RNA seq analysis and explain this?

For protein-protein interaction please give all details about the parameter that were set for STRING. Such as confidence level, algorithm, and all other parameters to generate the network.

Author Response

We would like to thank the reviewers as well for their valuable comments. Below is a detailed point-to-point response to each of the reviewers’ comments.

Reviewer 3

Specific Comments:

This manuscript uses the in vitro and in vivo approach to suggest the therapeutic potential of Ulixertinib-based treatment blastoma.

Strengths:

Well written manuscript
Data clearly shown in figures and images
Good level of interpretation

Limitations:

Sample size unknown in some places.

Thank you for your comment. We have revised accordingly.

Please provide the catalog and company name for all chemicals/kits/reagents

Thanks. We have revised accordingly.

Please provide more detailed information on patients-derived xenograph.

Thanks. We have revised accordingly.

It will be appropriate to describe the dosage and time of treatment in the material and method section. Please give a detailed description of how the experiment was done, what dosage was included, and what time point were considered for the study. It will help readers to reproduce the study for research purposes.

Thanks. We have revised accordingly.

How many colonies were captured for counting?

Thanks. We have re-made figure to show the number of colonies in the treated group. The following is the real number of colonies in each treatment group.

NGP

DMSO

IC50

5*IC50

SK-NBE(2)

DMSO

IC50

5*IC50

1

305

74

7

1

987

547

7

2

269

105

9

2

1475

354

13

3

274

135

13

3

894

489

19

Average

282.6667

104.6667

9.666667

Average

1118.667

463.3333

13

Stdev

19.50214

30.50137

3.05505

Stdev

312.0774

99.02693

6

SK-N-AS

DMSO

IC50

5*IC50

CHLA255

DMSO

IC50

5*IC50

1

269

102

11

1

687

14

2

2

287

69

8

2

748

0

5

3

304

88

14

3

1042

22

9

Average

286.6667

86.33333

11

Average

825.6667

12

5.333333

Stdev

17.50238

16.56301

3

Stdev

189.8166

11.13553

3.511885

For the immunoblotting section – Please include detailed information. What antibodies and concentrations were used with their catalog no and company name?

Thanks. We have revised accordingly.

Please acknowledge the STRING site or link for the protein-protein interaction study.

The link was added in material and methods section 2.10 and results section 3.4.

Please correct lines 205-206 for the FC approach. It should be more or less than 2.0 FC for significantly altered genes.

Thank you. We have revised it accordingly.

Figure 1B lower bar diagram - The X-axis is not clear for both bottom figures. Please modify this.

Thank you. Figure 1B has been revised based on your comments

Please provide the densitometric bar image for all the western blots.

Thank you. We have revised accordingly to show the density of each band in the western blots.

Please give details about RNA seq data. What time point was considered for the RNA seq analysis and explain this?

The details have been added in Materials and methods section 2.8.

For protein-protein interaction please give all details about the parameter that were set for STRING. Such as confidence level, algorithm, and all other parameters to generate the network.

 The details have been added in Materials and methods section 2.10 .
